# Quantifying yield losses from Bt resilience among maize cultivars in South Africa

Jesse Tack [1] ✉, Courtney F. Cooper[2], Lawton Lanier Nalley[2], Petronella Chaminuka[3], Safiah Maali[4], Erin E. Farmer[5] & Michael A. Gore [5]

Genetically modified crops have provided economic and social benefits since becoming commercially available. One of the most successful and widely used applications is the integration of genes from the soil bacterium *Bacillus thuringiensis* for protection against damaging pests. Here, we leverage a robust dataset of 85,133 field-trial maize observations spanning all major production regions in South Africa from 1980–2018 to estimate yield gains associated with the first wave of genetically modified cultivars and explore the potential dynamic erosion of these gains since resistance was reported among first wave of single gene *Bacillus thuringiensis* cultivars. Leveraging the cultivars commercial release year, we find that genetically modified yield gains increased dynamically from their initial introduction in 2000, peaking at approximately 0.55 MT/ha around 2006, after which they significantly eroded to near-zero by 2014. Interestingly, this erosion was followed by a dramatic rebound in gains, reaching an in-sample high of approximately 0.75 MT/ha.

Genetically modified (GM) crops have provided economic and benefits since becoming available for commercial production in the US in 1996[1,2]. The global economic gains from GM adoption is estimated at $224.9 billion USD distributed amongst more than 16 million farmers[3] via increased yields[2,4–7] and decreased pesticide costs as well as increased household income[8,9]. In addition to economic gains, GM crops have been found to have socioeconomic impacts that can improve producers' quality of life, such as environmental benefits and labor-saving technology[10,11]. Impoverished farmers in low-income countries have benefited the most from GM technology, where there are fewer options for pest management and crop vulnerability tends to be higher[1,12].

One of the most successful and widely used applications of GM technology globally is the integration of genes from the soil bacterium *Bacillus thuringiensis* (Bt) for protection against damaging pests. Producer and environmental benefits associated with the adoption of Bt crops have been well documented[13–18]. Improved insect control has resulted in increased yield and fewer insecticide applications, leading to cost savings and reductions in pesticide toxicity[15,16,19]. Another indirect impact of Bt maize adoption has been the reduction of

mycotoxin (e.g., fumonisin and aflatoxin) contamination, which has resulted in economic benefits and improvements in human health[20]. The improvements to human health stemming from mycotoxin reductions also disproportionately benefit low-income countries where fumonisin and aflatoxin levels are often high and maize serves as a staple food[21]. Currently, one of the biggest threats to these welfare gains is the evolution of resistance to Bt traits among pests and this is a very active area of scientific research[22–26].

Our focus here is on the productivity effects, and potential erosion thereof, associated with Bt maize in Africa where fall armyworm (FAW) (*Spodoptera frugiperda* (J.E. Smith) (Insecta: Lepidoptera: Noctuidae) currently contributes to significant crop loss where conventional breeding of tolerant crop cultivars and integrated pest management approaches have proven too slow to halt the spread of the emerging threat[27]. FAW was first reported in Africa in 2016 on maize plants in the rainforest zone of South-Western Nigeria and in maize fields at the International Institute of Tropical Agriculture and has since become established in many areas across the Western, Northern, and Central parts of the continent[28,29]. By January 2017, The South African Department of Agriculture, Forestry, and Fisheries

[1]Department of Agricultural Economics, Kansas State University, Manhattan, KS, USA. [2]Department of Agricultural Economics and Agribusiness, University of Arkansas, Fayetteville, AR, USA. [3]Agricultural Research Council, Pretoria, South Africa. [4]Agricultural Research Council - Grain Crops, Potchefstroom, South Africa. [5]Plant Breeding and Genetics Section, School of Integrative Plant Science, Cornell University, Ithaca, NY, USA. ✉e-mail: jtack@ksu.edu

(DAFF) confirmed the presence of FAW in Limpopo, Mpumalanga, North West and Free State provinces. By 2018, only ten of 54 African states and territories had not reported infestations of FAW. It is estimated that sub-Saharan Africa has 35 million hectares of maize grown by smallholders and that almost all of this is either infected or at risk of infection[30]. An ongoing challenge threatening the long-term effectiveness of Bt crops is their widespread adoption coupled with high selection pressure on target insects and low producer compliance with non-Bt structured refuge recommendations[31,32].

Bt resistance is particularly important to South Africa as 87% of its white grain maize crop was estimated to be GM[33] and is one of the largest GM maize producers in the world. In 2016, 74% of the country's total maize crop used herbicide-tolerant (HT) cultivars while 91% of the country's total maize crop used Bt cultivars[34]. Starting in the 2004/2005 season, South African maize producers noted severe maize stalk borer (*Busseola fusca*)-associated damage to Cry1Ab-expressing Bt maize[35]. The reduced pest control (>10% damaged plants) in the 2004/2005 season was attributed to the development of insect resistance to the Cry1Ab protein in MON810 maize[36,37]. By the 2007/2008 and 2008/2009 maize growing seasons, there was widespread resistance to MON810 in multiple areas, and by 2012 *B. fusca* populations with resistance to Cry1-Ab expressing maize were found throughout maize production in South Africa[37,38]. In response South Africa began enforcing refuge compliance and commercial released pyramided maize hybrids. From the 2012/2013 growing season onwards, the pyramided event MON89034, which expresses two different Cry proteins, Cry1A.105 and Cry2Ab2, was commercially adopted in South Africa in an attempt to address the problems that arose with resistance.

While many field level studies[37,39,40] have reported the presence of Bt-resistant insects across South Africa, to our knowledge there has not been a countrywide aggregated study that tested for the presence of reduced Bt yields across time due to resistance. Here, we leverage a robust dataset of 85,133 field-trial maize observations across 100+ locations that span all major production regions in South Africa from 1980–2018 to estimate yield gains associated with the first wave of GM cultivars and explore the potential dynamic erosion of these gains since Bt resistance was reported.

## Results

The results of our conventional, single Bt, single HT, and stacked (both Bt and HT) maize yield analysis are based on multi-year, multi-location maize trials conducted by the South African Agricultural Research Council's Grain Crops Institute. A randomized complete block design (RCBD) with multiple replications was used for the maize trials. Each locality was allocated its trial randomization, which differed annually. The trials assessed the adaptability of commercial genotypes entered by seed companies across a wide range of yield potentials. More details of the trials can be found in the Methods section below.

### Data and empirical approach

We focus specifically on dryland (rainfed) production and Provinces for which there is at least 30 years of trial data. The final dataset provides 85,133 observations spanning 36 years (1980–2018 with no data available in 1982, 1991, nor 2014), 104 locations, and 702 cultivars. Detailed information on the number of locations and cultivars by trial year are provided in the methods section. Fig. 1 provides box plots over years of the in-sample maize yields, demonstrating substantial cross-sectional and temporal variation in the data, as well as kernel density plots of the yields across cultivar type (conventional, single Bt, single HT, and stacked) that demonstrate the substantial overall variation in yields.

Table 1 provides summary statistics for maize yields across conventional and GM cultivars. Yields average 6.54 MT/ha across all observations, with a maximum of 23.96 MT/ha. While maize yields of zero occur in these trial data, they are rare (less than 0.02 percent). The average is lower among conventional cultivars at 5.99 MT/ha for all

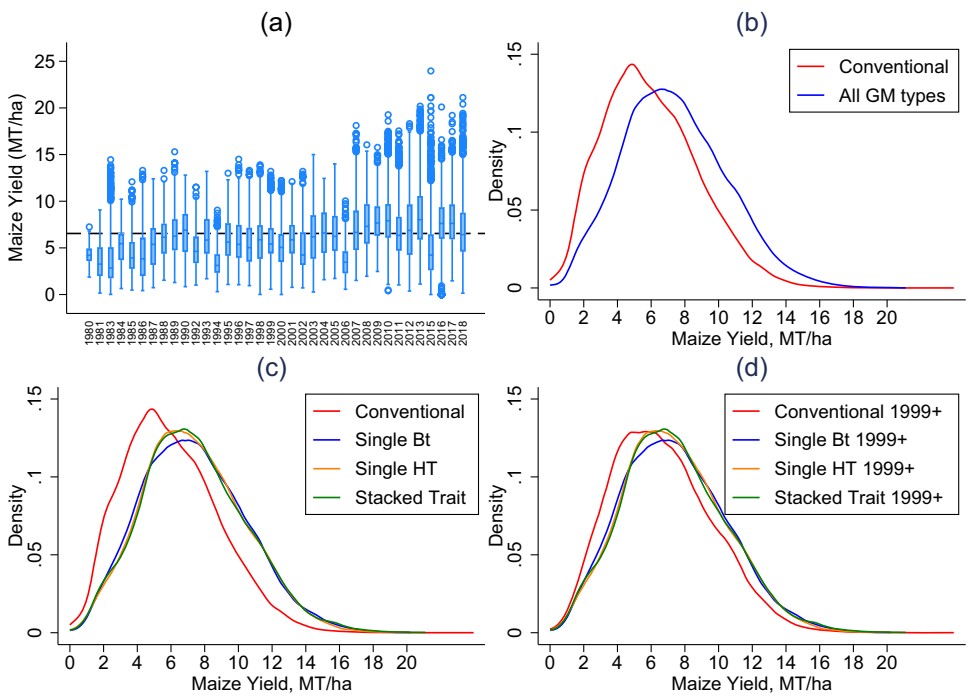

**Fig. 1 | Comparison of maize yields for South African trial data from 1980–2018. a** provides box plots of all cultivars and locations over time (box lines are 25th, 50th, and 75th percentiles; whiskers are lower/upper adjacent values; and circles are additional singular data points); (**b**) provides kernel density plots of conventional hybrid yields versus any genetically modified (GM) cultivar; (**c**) provides

similar information but breaks out GM across single-trait Bt, single-trait HT, and stacked-trait (both Bt and HT); (**d**) replicates (**c**) but only for years with conventional and GM overlap, i.e. 1999 to present. *N* = 85,133 in all panels. Source data are provided as a Source Data file (DataF1).

**Table 1 | Maize yield summary statistics**

| Yield | Obs | Avg | Std Dev | Min | Max |
|---|---|---|---|---|---|
| Yield (all trial years and cultivars) | 85,133 | 6.54 | 2.99 | 0.00 | 23.96 |
| Yield (conventional, all years) | 51,492 | 5.99 | 2.81 | 0.00 | 23.96 |
| Yield (conventional, 1999–2018) | 25,893 | 6.65 | 2.94 | 0.00 | 23.96 |
| Yield (GM cultivars, 1999–2018) | 33,641 | 7.39 | 3.06 | 0.00 | 21.11 |
| Yield (single Bt cultivars, 1999–2018) | 14,354 | 7.38 | 3.08 | 0.00 | 20.20 |
| Yield (single HT cultivars, 1999–2018) | 8807 | 7.37 | 3.01 | 0.00 | 20.10 |
| Yield (stacked cultivars, 1999–2018) | 10,480 | 7.42 | 3.08 | 0.00 | 21.11 |

Notes: Yields are reported in metric tonnes per hectare. 1999 is the year in which the GM cultivars first appeared in the trials. Source data are provided as a Source Data file (DataT1).

trial years. Since 1999, when GM cultivars first appear in the trials, conventional cultivars averaged 6.65 MT/ha while GM cultivars averaged 7.39 MT/ha. Breaking this out across GM traits, single Bt averages 7.38 MT/ha, single HT averages 7.37 MT/ha, and stacked (both Bt and HT) averages 7.42. Based on these summary statistics alone, GM cultivars are associated with an approximately 0.74 MT/ha yield gain relative to conventional, and the gains appear uniform across separate trait categories. This is also supported by Fig. 1C and D, which suggest that GM cultivars are associated with a rightward shift of the yield distribution. However, these comparisons fail to take into account potential biases associated with different trial locations (i.e., soil, climate) and years (i.e., weather, management practices).

Our empirical application leverages panel fixed effects models, which control for trial-location-by-year fixed effects. The location component of these controls for location-specific factors including soil quality, climate, and localized management practices, while the year component controls for changes in management practices over time in addition to common shocks associated with improved genetics, annual weather fluctuations, and pest/disease pressure. By crossing these fixed effects, note the use of "by" in the leading sentence above, we also are able to control for location-specific weather/pest/disease that vary over time. This is especially important in this context as the farm location at which the trials were conducted may or may not have been following refugia recommendations, which could in turn bias the estimated GM effect.

Our effect of interest is the yield gain associated with GM cultivars relative to conventional. The first set of results assumes a homogeneous effect of all-GM cultivars regardless of the underlying traits nor the timing of their availability. We sequentially increase the sample size to include more recent years to examine whether the average GM effect changes over time. The next set of results models this progression of the GM effect directly by interacting the GM variable with a cubic function of release year, where release year is defined as the year in which the cultivar was commercially released and is measured relative to the first trial year in which GM cultivars appeared in-trial (i.e., 1999, see Methods for more details). We considered other measures based on the trial year in which the cultivar appeared and linear, quadratic, and spline specifications of time but found that our preferred model outperformed them in an out-of-sample prediction exercise (results discussed in Methods). Using this cubic model, we also conduct split-sample analyses to directly compare the effects of single Bt, single HT, and stacked (both traits) cultivars to conventional, as well as a direct comparison of single Bt to single HT cultivars. Finally, the third set of results explores potential location heterogeneities of the dynamic GM effect across the five Provinces represented in the data.

**Homogeneous GM effect**

Our first set of results estimates the homogeneous yield gain associated with GM cultivars for sequentially rolling subsets of the data. Starting in 2005, the effect is estimated using Eq. (1) (see Methods) and

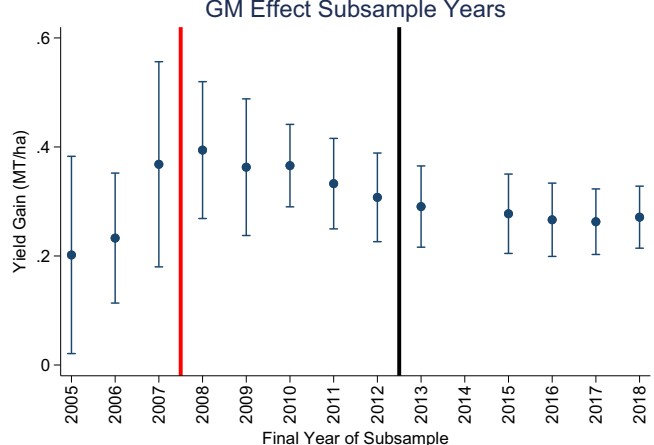

**Fig. 2 | Estimated effect of GM for sequential temporal subsets of the data.** Subsets are defined by the years 1980-X, where X is reported on the horizontal axis. Circles are parameter estimates and bars are 95% confidence intervals that are robust to spatial correlation. The vertical red line indicates when resistance to Bt was first reported while the black line indicates when the new generation of Bt cultivars were released. See Table SA1 for N in each subset. Source data are provided as a Source Data file (DataF2).

data from 1980–2005. We repeat this exercise for all years up to the final in-sample trial year in 2018, and the yield gain estimates are reported in Fig. 2 and Table SA1 (note all table and figure numbers that begin with an "SA" are provided in the supplementary information). Fig. 2 includes vertical lines demarcating the year in which Bt resistance was first reported (i.e., 2008) and when new cultivars were developed to combat this resistance (i.e., 2013). The gain in 2005 is estimated to be approximately 0.20 MT/ha ($p$ value < 0.05) and sequentially increases to a high of 0.39 MT/ha ($p$ value < 0.001) in 2008. Over the next few years, the estimate sequentially drops to 0.29 MT/ha ($p$ value < 0.001) in 2013, where it remains through the end of the sample.

Next, we consider splitting the sample across the years 2005–2007, 2008–2012, and 2013–2018 and re-estimate the model in Eq. (1). The breaks before the 2008 and 2013 seasons were chosen based on when resistance was becoming widespread and the release of the pyramided cultivars (see Introduction above). In each of these subsamples, we only include cultivars released during that period and focus exclusively on comparing conventional yields to those of (stacked) GM cultivars with both the Bt and HT traits. The estimated effects are reported in Figure SA1, and we see some important similarities and differences relative to the gains for the all-GM comparisons in Fig. 2: (i) the pre-2008 advantage of 0.64 MT/ha ($p$ value < 0.01) was much larger; (ii) the advantage substantially erodes after resistance is

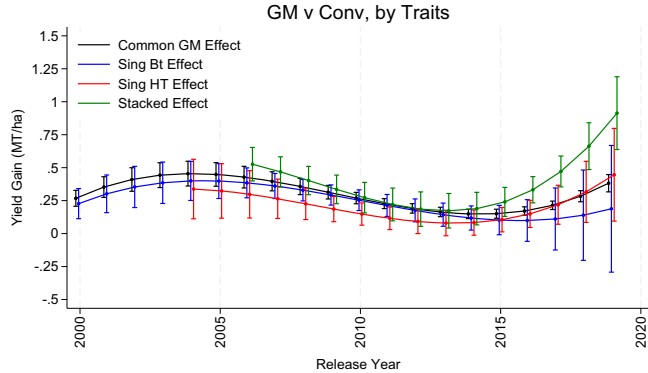

**Fig. 3 | Estimated GM effect as a function of the cultivars commercial release year, where the effect is relative to conventional.** We report effects across two approaches, the first assumes a common GM effect across traits while the other allows the effect to vary across single Bt, Single HT, and Stacked traits. Circles are parameter estimates and bars are 95% confidence intervals that are robust to spatial correlation. N = 85,132. Source data are provided as a Source Data file (DataF3).

reported; and (iii) the advantage improves when the new cultivars aimed at combating this resistance were developed.

The main takeaway from these results is that GM cultivars have exhibited a yield advantage relative to conventional. Still, this advantage begins eroding around the time that Bt resistance was first reported and then seems to improve at least for a subset of stacked cultivars.

## Dynamic heterogeneous GM effect

To estimate the dynamic gains more directly and move away from arbitrarily splitting the sample, we specify the GM effect as a cubic function of release year using Eq. (2) from the Methods. The advantage of this approach relative to sequential sub-sampling is that it does not combine the effects from previous trial years or releases, which can be thought of as a moving average of the effects, but instead estimates them directly for each release year based on an assumed functional form.

The next set of results correspond to this model and are reported in Fig. 3 and Table SA2. Column M2 of Table SA2 reports the estimated coefficients of the dynamic GM effect while still assuming a homogeneous (common) effect across different types of GM cultivars. The joint test of their statistical significance has a $p$ value < 0.001, and the year-by-year estimates are reported as a black line in Fig. 3. We find an increase in the estimated gains across release years up to 2004, which is associated with a yield gain of 0.45 MT/ha ($p$ value < 0.001), after which they decrease substantially until 2014/15. While still greater than zero at approximately 0.15 MT/ha ($p$ value < 0.001), the GM effect represents a near 70% reduction from its high in 2004. After this reduction, the yield gains turn upward and re-capture a substantial portion of the eroded gains compared to their previous high. The most recent GM effect for cultivars released in 2019 is 0.38 MT/ha ($p$ value < 0.001). Overall, there has been a substantial erosion of the GM effect, but some adaptive response has occurred that is restoring a large portion of this loss.

We also extend this model to allow the release-year effects to vary across trait groups, which are the colored lines in Fig. 3 and correspond to model M3 in Table SA2. Each of the release-year interactions are jointly statistically significant with $p$ values < 0.0001 for Single Bt and Stacked, and <0.05 for Single HT. Importantly, assessing the statistical significance of yield gains among one trait group versus another cannot be determined by comparing confidence intervals; so we take a more targeted approach based on pairwise comparisons. That is, to gain a better understanding of the mechanisms through which the

estimated GM yield gains may be eroding, we compare estimates of the gains using the following pairwise comparisons: (i) single Bt and conventional; (ii) stacked and conventional, (iii) single Bt and single HT; and (iv) single HT and conventional. For each comparison, we only keep cultivars in the data that correspond to one of the types, and then re-estimate model (2). For cases (i) and (ii), the gains are measured relative to conventional, while for (iii) and (iv) the gains are relative to single HT.

Equation (2) in the Methods section represents our preferred model and it has three parameters capturing the nonlinear (cubic) GM effect based on release year. These parameter estimates drive the shape of the main results reported in Fig. 4. The parameter estimates are reported in Table SA3 (columns M4-M7) and the p-values for a joint test of statistical significance are less than 0.001 for all but the single Bt vs single HT comparison.

The time-varying GM effect estimates for the pairwise comparisons are reported in Fig. 4. The single Bt trait was released in 2000 and represents the longest-running commercially available GM attribute within the trials. Looking at Fig. 4a, they provided significant yield gains relative to conventional from their release up to a peak of 0.39 MT/ha ($p$ value < 0.001) in 2005, after which they slowly eroded to an (essentially) zero yield advantage in 2019 (0.04 MT/ha, $p$ value = 0.848). Turning to Fig. 4b, stacked were released in 2006 and initially had a yield gain of 0.55 MT/ha ($p$ value < 0.001), which is substantially higher than its single Bt counterparts and likely attributed to the inclusion of the HT trait as it alone shows a yield enhancing effect relative to conventional (see Figure SA2). Interestingly, after 2005, the single Bt and stacked cultivars experienced significant erosion of the initial gains through 2014, after which the effects strongly diverged as the advantage of stacked-trait cultivars rebounds extensively up to an in-sample high of 0.76 MT/ha ($p$ value < 0.001) in the 2019 release year.

The overall pattern suggests that the cultivars with Bt traits were quite similar until recently, when a distinct change in breeding/release strategy occurred in response to eroded gains. This is also supported by a direct stacked and single Bt comparison provided in Figure SA3. One potential source of this divergence could be that the performance of cultivars with HT traits improved in the latter part of the sample, however, we see the exact same pattern of results when the Bt varieties (both single and stacked) are compared to single HT cultivars in Fig. 4c, d. This suggests that the recent increase in gains among stacked cultivars was not due to improved HT traits.

## Location specific dynamic GM effect

So far, we have only considered the heterogeneity of GM yield gains across release years, but the impacts could also vary across trial years and locations. We investigate this using two approaches: a split sample exercise where we re-estimate Eq. (2) separately for each province and a multilevel mixed model with random intercepts and slopes.

The results of the separate models for the Free State, Gauteng, KwaZulu-Natal, Mpumalanga, and North West provinces are reported in Figures SA3–SA7. In all cases, an initial yield gain of all Bt cultivars erodes substantially through the middle part of the sample, and in all but one case, the advantage rebounds to capture lost gains. The lone exception is the Gauteng province, where the advantage relative to single HT has eroded to zero.

The specification of the mixed effects model is identical to Eq. 1, except (i) the location-by-year fixed effects are replaced with random intercepts for location and year, and (ii) random slopes across release year, location, and trial year are included for the GM effect. This approach allows the effect of GM to vary across location, release year, and trial year. Figure SA8 summarizes these effects for the latter two dimensions, and the pattern of results is consistent with those reported above.

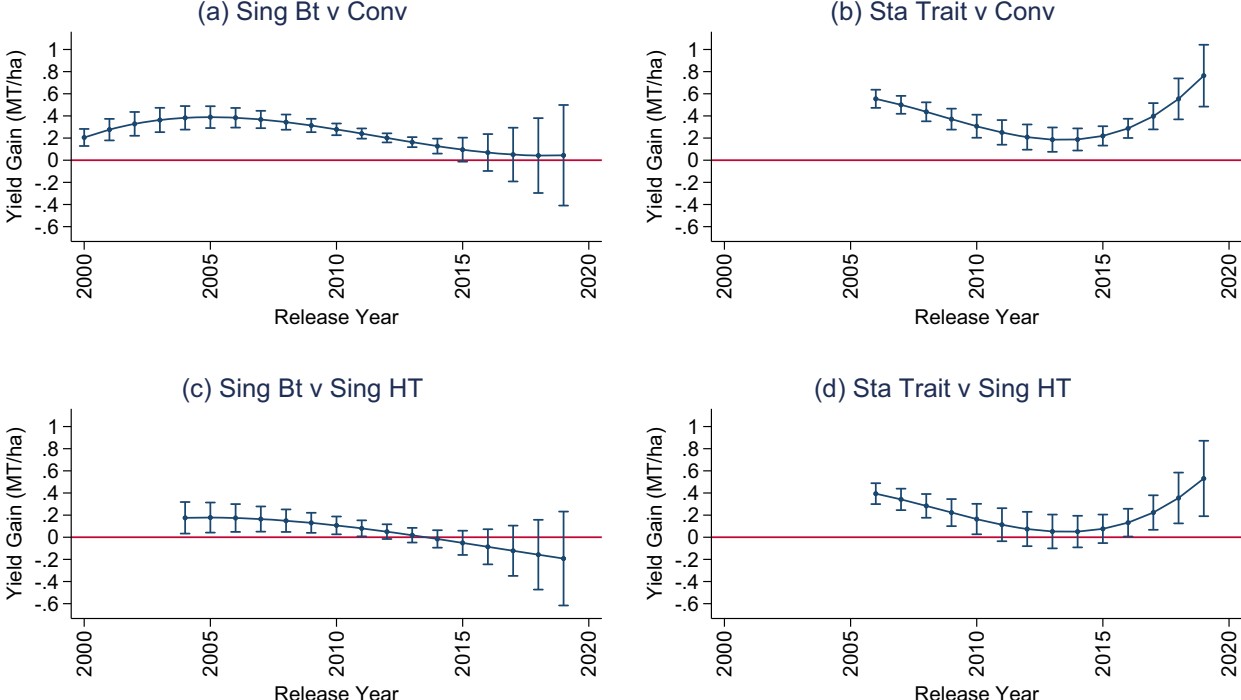

**Fig. 4 | Estimated GM effect as a function of the cultivars commercial release year. a** compares single-trait Bt cultivars to conventional, *N* = 65,845; (**b**) compares stacked cultivars to conventional (Conv), *N* = 61,971; (**c**) compares single-trait Bt cultivars to single-trait herbicide tolerant (HT) cultivars, *N* = 23,109; (**d**) compares stacked cultivars to single-trait herbicide tolerant (HT) cultivars, *N* = 19,266. Circles are parameter estimates and bars are 95% confidence intervals that are robust to spatial correlation. Source data are provided as a Source Data file (DataF4).

## Cumulative loss in food rations

Like many Southern African countries, South African consumers only eat white maize, with yellow maize being grown exclusively for livestock feed. As such, we demonstrate the economic significance of the GM yield gain erosion to South African maize consumers following a similar approach as Ala-Kokko et al.[33] GM yield gain estimates are combined with data on the total (national) area sown to white maize and per-capita consumption data (one yearly maize ration) to estimate the total loss in food rations between 2005 and 2018. Over this period the total cumulative losses are over 35 million white maize food rations, averaging approximately 2.3 million per year with a high of over 4 million in 2012/2013 (Table SA4 and Figure SA9). Importantly, these loss estimates were not due to the use of GM maize but rather the erosion of a yield premium relative to conventional maize due to resistance.

## Discussion

This study presents the first results investigating the spatial and temporal dynamics of GM crop yields relative to conventionally bred cultivars at a country level. Taken as a whole, our results are consistent with an initially large yield advantage of GM cultivars attributable to both Bt and herbicide tolerance traits, a period of resistance to Bt that eroded a large portion of this advantage, and then, more recently, a new line of Bt traits that were only incorporated into stacked cultivars. The degradation of efficacy of single Bt cultivars has broad implications for maize producers in South Africa and globally, as the well-documented benefits associated with GM crops are likely to be lost, including higher household income due to yield premiums[10], labor savings[41], and non-reliance on pesticides[42].

This erosion could be attributed to several factors, such as improved germplasm targeted in cultivars with stacked traits (Bt and

HT), a shift in breeding funding towards conventional cultivars as consumer pressure mounts against GM crops, insect resistance to Bt, or a paradigm shift in breeding trait outcomes such as drought tolerance. However, given how the timing of the first reported field resistance aligns with our estimated decline in Bt yields, coupled with how the timing of the addition of stacked Bt proteins (Cry1A.105 and Cry2Ab2) aligns with our estimates of increasing yields, resistance could be driving these yield swings. While we cannot directly test this resistance hypothesis due to a lack of pest population data at the scale of this study, any competing hypothesis would have to explain why the GM yield premium reduced for both single Bt and stacked-trait cultivars while only bouncing back for the stacked-trait cultivars. While field-level cases demonstrating the presence of resistance to Bt crops have been widely established, it is unknown how resistance may impact yield premiums across regions at a national scale.

Genetically modified crops like Bt maize have provided opportunities for crop improvement beyond the scope of what is possible with conventional breeding methods. Advancements in transgenics made it possible to move genes from other species into agronomically important crops, expanding the available set of phenotypes beyond their own natural variation. More specifically, Agrobacterium-mediated transformation of maize was achieved in the 1990s[43], paving the way for Bt and HT hybrid maize lines, which are now ubiquitous in South Africa. However, this technology has barriers and associated challenges, as transgenics rely on the discovery of the desired trait in another species and the isolation of its causal gene. In addition, even with advancements in maize for leaf transformation[44] and DNA-free genome editing[45], there still exists the challenge of rapidly developing new cultivars to keep pace with evolving resistance. A previous study, which surveyed 6 major seed companies, including Bayer, Syngenta, and the parent companies which would later lead to Corteva

AgriScience, reported that the pipeline from discovery to market takes about 13 years and $130 million, on average, in large part due to extensive testing and regulatory frameworks[46].

In addition to the challenges, time, and cost associated with developing new GM cultivars, the evolution of insect resistance is a widespread and prevalent threat to their efficacy once implemented. Recent research[47] summarized 73 field-level case studies of Bt crops from 7 different countries, reporting an increase in the number of cases of insect resistance from 3 in 2005 to 26 in 2020. It is also worth noting that, for every case, the time to resistance was equal to or less than the 13-year average needed to develop a new GM seed, with resistance developing after 6.6 years, on average[47]. This average is also consistent with the timing to potential resistance seen in the results above for South Africa.

In the case of insect susceptibility to Bt Cry proteins, the refuge strategy remains the most prevalent practice to prevent widespread resistance. Several field-level studies indicate the effectiveness of these strategies, findings that are not specific to the crop, region, insect, or specific Bt protein[31,47]. However, it was estimated that in 1998, one year after the commercial release of MON810 (the first commercial release of a Bt cultivar in South Africa), only 7.7% of producers that planted MON810 actually planted the refuge they were legally obligated to plant[35].

If insect resistance becomes fixed within a population, new or different control methods must be introduced to manage pests and limit damage. Synthetic insecticides, for example, suffer from similar challenges if not managed correctly, are expensive, and may have other negative externalities, particularly in terms of their environmental impact[48]. Additional strategies of control include pyramiding Bt Cry proteins, which was implemented in South Africa in 2011, and RNA interference via transgenics[49]. However, each of these suffer from many of the limitations discussed previously, particularly those associated with transgenics, as well as their own unique challenges.

Overall, it is unclear if the development of new approaches or the introduction of additional Bt Cry proteins can outpace the evolution of resistant insects. Given the current reliance on Bt Cry proteins to confer insect resistance, effective management strategies, such as the use of refuges, are necessary to delay or prevent widespread resistance, erosion of yield premiums, and loss of the economic, social, and environmental benefits of GM crops[26]. The results of this study quantify the yield premiums that GM technology can generate but also highlight how quickly those premiums can dissipate across national production systems if holistic management practices are not implemented. This is especially important given previous research documenting that concurrent use of transgenic plants expressing a single and two Bt genes speeds insect adaptation to pyramided plants[50].

## Methods

Maize trial data for dryland production are combined across all locations in the Free State, Gauteng, KwaZulu-Natal, Mpumalanga, and North West Provinces. Data was also available for the other provinces but was not included because those provinces did not provide rich enough information to measure the dynamics of the GM effects as they contained less than 30 years of information. Among the full data that contained all provinces, the included provinces represented approximately 92 percent of all yield observations (Free State 20%, Gauteng 10%, KwaZulu-Natal 15%, Mpumalanga 19%, and North West 28%). The omitted provinces were Eastern Cape (2%), Limpopo (3%), Northern Cape (1%), and Western Cape (2%). The final data for the five represented provinces contained 85,133 observations spanning 36 years (1980–2018 with no data available in 1982, 1991, nor 2014), 104 locations, and 702 cultivars. These data are made publicly available through the publisher's website and were previously used to estimate the homogenous effects of GM yields in South Africa[51].

We define a "trial" by unique location-year combinations, of which there are 934 in the data. Table SA5 provides the characteristics of these field trials by year. Here and throughout, year is defined by the calendar year the maize was planted. The average number of trials across years is 26, with a low of 5 and a high of 36. The data begins in 1980, but GM cultivars do not appear until 1999, after which representation was initially less than five percent of annual observations for the first five years, rising quickly to approximately 20% in 2005 and steadily increasing thereafter with a high of approximately 78% in 2013. From 2010 to 2018, GM cultivars averaged approximately 70% of field trial observations. Bt was the first trait introduced in 1999, with herbicide tolerant (HT) traits first appearing in 2003, followed shortly thereafter by stacked cultivars (both Bt and HT) in 2005. From 1999 to 2018, Single Bt cultivars represented approximately 24% of all trial observations (including conventional), Single HT 15%, and stacked 18%; however, the general trend since 2010 has been a reduction of single Bt representation coupled with an increase in both HT and stacked representation.

A randomized complete block design (RCBD) with three replications (blocks) was used throughout and each locality was allocated a new trial randomization each year. We considered including the replicate number as a control variable but it lacked statistical significance ($p$ value $= 0.728$) and the estimate of the GM effect was numerically equivalent to four significant digits compared to the one reported (Table SA2, M1). Each locality was allocated its own trial randomization that differs annually. The same 50 genotypes were used in all the trials and consisted of genotypes entered by the seed companies involved. Seed company entries are in order of priority. Where too many entries were received the final genotype choice was made through negotiations with each seed company. Stands of 40 plants per netto plot were prescribed throughout whereas the choice of plant population (plants/ha), row widths and spacing was left to the discretion of the co-workers. A number of 22 plants in each of two plant rows per gross plot were recommended for row widths less than 1.5 m, while a number of 42 plants in a single plant row per gross plot was recommended for row widths of 1.5 m and greater. Border rows were only prescribed for the perimeter of the trial and no border rows were required between plots. One boundary plant had to be removed from each end of the plant row at harvest. Fertilizer applications were not prescribed but applied according to soil fertility and the maize yield potential of that area. Seed dressing is widely used in South Africa and in the trial data. The same rate is applied regardless of the technology throughout the trials. Trials were combine-harvested when the average grain moisture was 20% or lower, with a target of 13%. All cultivars in the trials are hybrids.

Table SA6 provides the cultivar characteristics within the field trials over time. The number of cultivars tested in a given year averaged 57, with a low of 48 and a high of 86. There were only two GM cultivars in 1999 and the number stayed below ten until 2005, the first year all three GM types were represented in the data. The number of GM cultivars reached a high of 68 in 2013. In total, there are 205 unique GM cultivars in the data, 84 of which are single Bt and 57 are single HT. Among the conventional cultivars, some have appeared in as many as 370 trials and as few as 5, averaging 68 (Figure SA10). Among the single Bt, single HT, and stacked GM cultivars, the average number of trials is 60 (high of 301), 49 (high of 210), and 47 (high of 329).

We also investigate whether the field trials represent modern genetics as they are sequentially conducted over time and find that they indeed do. We measure "release year" by the year in which the cultivar was commercially released, and Figure SA11 provides box plots of the in-sample representation of release years over time broken out by cultivar type. The plots suggest that the trials represent modern genetics as older cultivars are replaced with more recent ones regularly. This release year definition is the same as used in our preferred empirical specification below.

Our statistical model for estimating the yield gains associated with GM cultivars leverages field-trial-location-by-year fixed effects to control for omitted variable bias from unobserved factors that could be correlated with the appearance (or performance) of GM cultivars in the trial. The location component controls for all time-invariant factors at the location level (e.g., climate, soil quality), while the year component effects control for pest pressure, weather shocks, and non-GM technological improvements over time that are common across all cultivars. These include management improvements (e.g., increased fertilizer), widespread droughts or heat waves, "stock" germplasm improvements that are common to both conventional and GM cultivars, and evolutions of pest pressure either through a change in population or resistance to pest management strategies that include GM-specific traits. By crossing these fixed effects, note the use of "by" in the leading sentence above; we are also able to control for any of the above confounders at the location-year level. This is especially important in this context as the farm location at which the trials were conducted may or may not have been following refugia recommendations, which could in turn bias the estimated GM effect.

Another potential concern that the fixed effects control for is the strategic selection of cultivars into certain locations based on local growing conditions (e.g., climate, soil quality, pest/disease pressures). Furthermore, the random block design of the trials that include both conventional and GM cultivars side by side will likely control for other forms of endogeneity not mentioned above. Beginning in 2005, every trial in the sample included both conventional and GM cultivars, with the percentage of the latter ranging between 16–83% and averaging 58%.

Denoting the yield for cultivar $i$ at field-trial location $l$ in year $t$ by $y_{ilt}$, the first regression model we consider assumes a homogeneous GM effect and is given by

$$y_{ilt} = \alpha_{l,t} + \beta GM_i + \varepsilon_{ilt} \tag{1}$$

where the $\alpha$ are the fixed effects, $GM_i$ is a dummy variable equal to one for GM cultivars, and $\beta$ is the parameter of interest capturing the yield gain associated with GM. Multiple replicates are included in the sample however we omit its notation in the model for convenience. We cluster the standard errors by year to allow for widespread spatial correlation of the error term within and across all trials.

The next model we consider allows the GM yield effect to evolve dynamically based on the release year of the cultivar. Since GM cultivars first appeared in the 1999 trial year, *release* takes on a value of 0 before 1999, 1 in 1999, 2 in 2000, and so on, up to a value of 20 in 2018. The parameter $\beta$ from Eq. (1) is now generalized as a cubic function of *release*, with the resulting specification being:

$$y_{ilt} = \alpha_{l,t} + \left( \beta_1 release_i + \beta_2 release_i^2 + \beta_3 release_i^3 \right) GM_i + \varepsilon_{ilt}. \tag{2}$$

We considered several other specifications for the dynamic GM effect, including replacing release-year with the year in which the trial was conducted and alternative temporal specifications, including linear, quadratic, and spline specifications. In all, we tested ten different models, the three polynomials and two cubic splines with varying knots of three and five, each of which was combined with the two measures of time. These models were tested for prediction accuracy using an out-of-sample bootstrapping procedure that randomly sampled 80% of the observations to train the model. The left-out 20% of observations were used to measure prediction accuracy, which we report as the percentage reduction in root mean squared error (rmse) relative to the baseline model with a homogenous (time-invariant) GM effect, i.e., Eq. (1) above. The sampling procedure was conducted 1000 times and the results are reported in Table SA7. We select as the preferred model the one with the highest prediction accuracy (largest rmse reduction), which is the cubic specification in release year.

To the extent that we have a single overarching model, it is given by Eq. (2) in the methods. This is the model used to estimate our main findings which are reported in Fig. 4, where each panel reports estimated yield gains for each of four subsamples. By first subsampling the data and then estimating the pairwise relationship, the location-by-year fixed effects are unique to that pairing. Since these are the main control variables for our identification approach this represents a more robust approach than pooling the data together and estimating Eq. (2) with different indicators for the GM categories. This same logic applies to the province-specific approach; by splitting the sample and then estimating the effects we are leveraging a more robust model. It is also worth noting that we do compare our results to a mixed model, which is arguably the best possible single (overarching) model approach one could consider in this context, and the results are consistent with our preferred approach.

### Ethics and inclusion statement
The research was conducted in collaboration with local partners and researchers, who were involved throughout the study, with roles and responsibilities agreed upon in advance. The project is locally relevant, jointly defined with local collaborators, and does not raise ethical, legal, or regulatory concerns in the research setting. No human, animal, environmental, or biosafety risks were identified, and the study does not pose risks of stigmatization, discrimination, or harm to participants or researchers. Finally, relevant local and regional research was appropriately acknowledged and cited.

### Reporting summary
Further information on research design is available in the Nature Portfolio Reporting Summary linked to this article.

## Data availability
Source data for the analysis is named "RegData". All figures and tables are produced from these data, and individual source files for each of the main figures and tables are also provided. Data for Fig. 1 is named DataF1; data for Fig. 2 is named DataF2; data for Fig. 3 is named DataF3; data for Fig. 4 is named DataF4; and data for Table 1 is named DataT1. Source data are provided with this paper.

## Code availability
Code for replicating the main results and generating the source data for each of the main figures and tables is provided as Supplementary Code 1.

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

## Acknowledgements

We acknowledge and thank the South Africa Agricultural Research Council (ARC), Grain Crops Campus for the data used in this study. All opinions expressed in this paper are the authors' and do not necessarily reflect the policies and views of ARC.

## Author contributions

J.T., C.F.C., L.L.N., E.E.F., and M.A.G. either worked directly on the data analysis and/or provided feedback on the methodology. P.C. and S.M. provided the data and the information related to its collection. All co-authors contributed to the writing of the manuscript.

## Competing interests

The authors declare no competing interests.
