## [Peer Review file · Nature Communications]

Quantifying Yield Losses from Bt Resilience among Maize Cultivars in South Africa

Corresponding Author: Dr Jesse Tack

Version 0:

Reviewer comments:

Reviewer #1

(Remarks to the Author)

I am an applied entomologist with expertise in field crops, and not an economist, and I will approach the review with this expertise.

This is a study that puts together an impressive database. The authors pull in multiple data points across South Africa corn production to look at yield differences among cultivars with conventional (non-GM), Bt traits, and herbicide tolerance (HT). I think the dataset is interesting and, as analyzed, the results are interesting. But I think the manuscript goes too far without more information. I would like to believe that the yield differences are due to GM traits, as the authors claim, but remain unconvinced that the differences are not due to genetic gain as a result in improved breeding toward cultivars with GM traits. Moreover, the results could result from pest suppression, a well-documented phenomenon due to widespread use of Bt traits, rather than yield gains due to Bt crops.

Lines 237 to 238 state what I think is going on. The authors admit that their results may be due to “improved germplasm targeted in cultivars with stacked traits”. Companies charge more for maize hybrids with GM traits compared to those that do not have GM traits. Therefore, it is in the company interest to increase breeding effort toward these hybrids that they can charge more for. This has been documented in the US and is thought to be one of the reasons why growers do not plant the required 20% non-Bt refuge.

The description in lines 358 to 363 and portrayed in figure SA11 is an attempt to sidestep this argument, although it's not overtly stated. The problem is that we still don't know how much breeding effort for yield gain went into Bt or non-Bt hybrids. There is no detail on what the hybrids are. In the US, both Bt and non-Bt hybrids developed from larger seed companies (Dekalb and Pioneer, for example) are often sold to smaller seed companies and re-labeled for sale. Were the hybrids that were tested in this study simply relabeled from year to year or were they unique? There isn't enough detail for me to judge this.

Further supporting this, the explanation that reports of field resistance timing (line 241) with a decline in Bt efficacy are unconvincing without any measurement of insect feeding in non-Bt or Bt hybrids. The authors report “widespread resistance” by 2007-2009 and the introduction of Cry1A.105 + Cry2Ab2 in 2012/2013, and the yield upswing in Bt versus non-Bt, as well as single Bt versus pyramided Bt's (Figs. 2-4) make this plausible. However, pest suppression could explain the difference in yield rather than resistance. As more and more Bt maize was planted in South Africa, populations could be suppressed by Bt. This would mean that the Bt yield advantage would decrease over time relative to the non-Bt hybrids. The yield upswing could be due to many other factors- improved breeding for yield in Bt hybrids, the introduction of an invasive insect, natural pest population increases, etc.

The lack of pest population measurements is concerning for another reason. Let's accept the premise that the results are due to resistance. Often when resistance evolves, the resistance is localized across the landscape, rather than being evenly spread across the landscape. The approach that the authors used could miss this effect if the trials were located in areas where resistance was either high or low. Line 369 states that the model included location by year as a component to eliminate pest pressure. But it's unclear whether location could have had outsized effects that the statistical model might not have removed.

Finally, as mentioned previously, there is too little information to make this experiment repeatable. For example, the only detail we are given on the field designs is from lines 332 to 347. This explanation is not sufficient. The authors refer to “cultivars”, which makes me wonder if all the non-Bt cultivars were hybrids, rather than open pollinated varieties. It would not be fair to compare these without specifying this.

As another example of how detail is important, in the US, all field corn hybrids are overtreated with a seed dressing of insecticide prior to sale. However, the rates of these insecticides tend to be higher on Bt hybrids compared to non-Bt hybrids. Because of this, any potential yield differences could be due to the soil insect pest complex and not Bt.

A couple specific questions:

Line 138- is this what lines 405 and 406 are describing. Different “release” points by year? What is the justification for breaking at these points?

104 locations (line 312) times 36 years is a ton of degrees of freedom. I would suggest a statistician look at this. Were locations combined?

(Remarks on code availability)

Reviewer #2

(Remarks to the Author)

The authors have gathered more than 30 years of field-trial data from across South Africa to address the question of whether the effect of genetically-modified (GM) cultivars on yield changes over time and, if so, how. Three different categories of GM cultivars are considered: the soil bacterium *Bacillus thuringiensis* (Bt) cultivars, herbicide-tolerant (HT) cultivars, and cultivars that are both Bt and HT. The yields of the GM cultivars were compared to that of conventional non-GM cultivars. Various models were developed to quantify the effect of GM cultivars on yield. This review focuses on the statistics used in the analysis.

The overall design of the study needs clarification. The authors specify that a “randomized complete block design (RCBD) with multiple replications was used throughout and each locality was allocated its trial randomization” (lines 328-329). What is meant by “multiple replications”? Is it the number of blocks within an RCBD at a given location, the number of RCBDs within a location, the number of RCBDs across locations, or the number of RCBDs both within and across locations? This would help the reader understand what a replicate number represents and perhaps why it had no significant effect. If the replicates are RCBDs within a location, was the randomization the same for all RCBDs?

How was y_{ilt} , the yield for cultivar i at field-trial location l in year t , computed? Was it the yield for each plot, multiple observations for each cultivar within the RCBD, the least squares mean from the individual RCBD analyses, or the average of the observed cultivar i yields? (These last two differ if any missing values are present within the RCBD.)

In general, one should determine whether an effect is fixed or random and let that determine the model fit instead of fitting effects as both fixed and random. Was a mixed effects model fit? That would require either the intercept or the GMs to be fixed and the other random; otherwise, if both are random, it is a random effects model. Note: The slopes are not random for the mixed effects model; the GMs are random. Given the study design, the assumption that GMs are random seems most reasonable.

For the primary modeling effort, the three different types of GM cultivars are grouped together to determine the impact of GM cultivars on yield. The effect of the GM cultivar differs among the Bt, HT, and both Bt and HT cultivars. The factorial nature of the cultivar types could be reflected by considering Bt and HT as indicator variables with a value of 1 if, respectively, Bt or HT is present. For the conventional cultivar, both indicator variables would be 0. By collapsing them into one category, the effect of GM depends on the effect of each of the three GM types and the proportion of times each occurs in the dataset for a specific year affects the estimates. Thus, a change in the effect of GM could be simply due to a higher proportion of one type of the GM cultivars occurring in the test data in a given year and that proportion may not be representative of the population of interest (maize grown in South Africa).

The authors have considered several follow-on analyses. To the extent possible, it would be better to have an over-arching model that would allow the desired comparisons. As an example, all types of GM cultivars could be part of the initial model. Another example would be to include a categorical variable for province and its interaction with GM.

A large number of tests are being conducted. Some correction should be made as the type I error rate is not well controlled as it stands. Prediction bands, which are a little wider than confidence bands, for the fitted curves are more appropriate than showing individual confidence limits as in Figures 3 and 4. Since the curve is significantly different from $y=0$ and the prediction bands would probably lie fully above the zero line, one could produce an estimate of the effect of GM for any point in time. The model becomes more complex, but a deeper understanding of the system is gained.

The authors have undoubtedly spent an enormous amount of time obtaining, cleaning, and analyzing these data. The results

are potentially interesting. It would help the reader if some of the statistical work, especially the design, was better described and more care was taken with some statistics.

(Remarks on code availability)

Reviewer #3

(Remarks to the Author)

This is a very interesting manuscript with some high quality data and an important story to tell to a wide audience.

To summarise the key results these are:

1. Bt maize cultivars led to yield increases compared to conventional cultivars
2. the yield advantage started to disappear once insect populations started to develop resistance to the single Bt gene (protein)
3. the yield advantage was reinstated (and possibly better) when cultivars with two pyramided (and crucially different to the first single Bt Cry1Ab gene and protein)

However there are some significant improvements to be made to include the clarity of the interpretation for the reader.

I do not believe the title is very clear in summarising the findings. It does seem to imply that Bt led to yield losses, whereas this was not the case.

Perhaps more engaging titles could be:

Two gene Bt maize cultivars were more resilient than single gene Bt cultivars in South Africa

or

Single gene Bt maize cultivars lacked resilience to insect losses in South Africa

or

Improving the sustainability of maize yield with two gene Bt cultivars in South Africa

Indeed the Abstract has some lack of clarity and should explicitly state that single gene Bt cultivars were released initially and failed because of "resistance in the insect populations". Indeed perhaps it could be concluded that the race to deploy single gene Bt was probably a mistake. And the deployment of 2 gene Bt also necessitated the use of 2 new and different genes. Indeed this had been previously widely predicted (for example see Zhao et 2005, PNAS - a team from Cornell).

Another issue that is not discussed is the returns to farmers. Higher yields would increase farm profitability. In addition, if insecticides were being used to control insects in conventional fields this would mean a higher input cost and overall a considerably lower profit margin.

This may not be easy for the authors to provide data but there could be some consideration as to the relative losses of maize to different insect pests and how that changed over time eg from *Busseola* to FAW? Did FAW have a large impact on conventional yields.

There are also some low yielding years in the mix (eg 2006) - was this due to insect pressure or a biotic stress?

Some minor comments"

line 18 "less insecticide application" should be "fewer"

line 219-223 should be explicit that the loss of over 35 million food rations was due to poor resistance management and not Bt deployment per se

Supp Figure SA11 - what is the significance of the horizontal red line on each graph

Ian Godwin

(Remarks on code availability)

Version 1:

Reviewer comments:

Reviewer #1

(Remarks to the Author)

This is my second time reviewing this manuscript. I appreciate the author's thoughtful responses to my concern, especially regarding pest suppression. However, I still think the authors are putting too much confidence in pest resistance as an explanation rather than breeding.

Figure 4 could simply tell a breeding story. Let's assume that companies put most of their breeding efforts into their most "elite" hybrids that they can charge a premium price for. In that case, corn breeders released hybrids with single Bt traits, with a yield advantage due to breeding that eroded over time. This yield advantage was apparent compared to both conventional corn hybrids (Fig. 4a) and single HT corn hybrids (Fig. 4c).

Breeders then stopped putting so much effort into these single Bt trait hybrids when stacked Bt trait hybrids were released (Figs. 4b and 4d). Why the upswing in yield? Breeding could explain it all to me.

I'm not refuting the author's claim that we can explain the dips and upswings in yield could be due to resistance. I only think they should make the point that breeding is a good alternative hypothesis that's not mutually exclusive.

(Remarks on code availability)

Reviewer #2

(Remarks to the Author)

Review of the revised manuscript "Quantifying Yield Losses from Bt Resilience among Maize Cultivars in South Africa" By Tack, et al.

As Reviewer 2 for the original manuscript, this reviewer will continue to focus on the statistics used in the analysis.

First, the authors have shown careful consideration of each of the comments, at times revising the manuscript and, at other times, explaining why they did not make changes in response to a comment. Thank you for making that effort.

In this version, it is evident that the results are robust to the choice of model, which adds confidence to the conclusions being drawn.

Apparently, one comment in the original review was not fully explained. It is the following:

"For the primary modeling effort, the three different types of GM cultivars are grouped together to determine the impact of GM cultivars on yield. The effect of the GM cultivar differs among the Bt, HT, and both Bt and HT cultivars. The factorial nature of the cultivar types could be reflected by considering Bt and HT as indicator variables with a value of 1 if, respectively, Bt or HT is present. For the conventional cultivar, both indicator variables would be 0. By collapsing them into one category, the effect of GM depends on the effect of each of the three GM types and the proportion of times each occurs in the dataset for a specific year affects the estimates. Thus, a change in the effect of GM could be simply due to a higher proportion of one type of the GM cultivars occurring in the test data in a given year and that proportion may not be representative of the population of interest (maize grown in South Africa)."

The authors assumed that the above comment was referring to Figure 4. In fact, Figure 4 illustrates that GM and time interact; that is, the differences among the different types of GM (single-trait Bt, single-trait Ht and stacked trait) change over time. Figure 3 depicts the main effect of GM, which is an estimated average of yield gains of GMs released in a given year and in the study; it is not clear whether all new releases are included each year. Because of the interaction, the yield gain for each year is not representative of any of the types of GMs when more than one type is present (see Figure 4) and is heavily dependent on the cultivars included in the study, which may not be reflective of the cultivars used by farmers. Although the authors note that the model and resulting Figure 3 are based on the assumption of "a homogeneous effect across different types of GM cultivars" and then explore the assumption, their interpretation appears to draw inference to the broader population, which is not homogeneous. In this case, the proportion of cultivars of each type included in the study has a substantial impact on the estimated yield gain simply because the yield gain varies with type. Thus, the recommendation was made to consider a model with type of GM (conventional (no GM), single-trait Bt, single-trait Ht and stacked trait) instead of only considering presence/absence of some type of GM in model 2. Due to interaction, this would lead to separate curves for the yield gain from each type of GM in Figure 3 and would eliminate the need for Figure 4.

Below are some further suggestions, none of which are major:

1. Beginning at the end of line 164, "the yield effect" should be "the GM effect"
2. Beginning at the end of line 181, "less than 0.000" should be "less than 0.001" (It is not possible to have a p-value less than 0.)

3. Line 343: Since there are three blocks (replicates) for each RCBD and a new randomization was conducted for each year-location combination, clarity could be added by changing “with multiple replications was used throughout and each locality was allocated its trial randomization” to “with three replications (blocks) was used throughout and each locality was allocated a new trial randomization each year”

4. For all graphs of yield gains (most graphs), the y-axis was labeled as “Yield (MT/ha).” This should be changed to “Yield Gain (MT/ha).”

5. In Figure 4, the title indicates that plot (d) depicts the yield gain of single-trait HT vs conventional, but the title above the plot says that it is the yield gain of stacked trait vs single-trait HT. This should be clarified.

Again, the authors have conducted a thoughtful analysis of a large, interesting dataset.

(Remarks on code availability)

Version 2:

Reviewer comments:

Reviewer #2

(Remarks to the Author)

As with the two earlier versions of this paper, this reviewer will focus on the statistical aspects. After the last review, one item remained unresolved, which was also present in the original review.

“Thus, the recommendation was made to consider a model with type of GM (conventional (no GM), single-trait Bt, single-trait Ht and stacked trait) instead of only considering presence/absence of some type of GM in model 2. Due to interaction, this would lead to separate curves for the yield gain from each type of GM in Figure 3 and would eliminate the need for Figure 4.”

The authors prefer to use Figure 3 to motivate the graphs in Figure 4. This does not address whether the interaction between GM and time is significant. That is, are the differences in the curves real or could they be attributed to error. To address the question properly, the appropriate model needs to be fit to the data. If there is no interaction, Figure 3 suffices unless the authors want to suggest that the power was insufficient to detect the interactions visible in Figure 4. If interactions are significant, then Figure 3 should be dropped or used to motivate fitting the appropriate model. Ideally, the model with the potential interaction should have been fit initially as it was important to the questions that the authors were asking.

(Remarks on code availability)

Version 3:

Reviewer comments:

Reviewer #2

(Remarks to the Author)

Thank you for considering the remaining concern and revising the paper to account for it. You have done a nice job of addressing my concern. This is a strong paper.

(Remarks on code availability)

Reviewer #1 (Remarks to the Author):

I am an applied entomologist with expertise in field crops, and not an economist, and I will approach the review with this expertise.

Thank you very much for the feedback, you raise some very important points that we address below and the manuscript is much improved as a result. Please note that we are providing a marked-up version of the manuscript that tracked all changes as well as a “clean” version that accepts all those changes. The line references in the replies below apply to the clean version.

This is a study that puts together an impressive database. The authors pull in multiple data points across South Africa corn production to look at yield differences among cultivars with conventional (non-GM), Bt traits, and herbicide tolerance (HT). I think the dataset is interesting and, as analyzed, the results are interesting. But I think the manuscript goes too far without more information. I would like to believe that the yield differences are due to GM traits, as the authors claim, but remain unconvinced that the differences are not due to genetic gain as a result in improved breeding toward cultivars with GM traits. Moreover, the results could result from pest suppression, a well-documented phenomenon due to widespread use of Bt traits, rather than yield gains due to Bt crops.

Thank you very much for your feedback and stressing the importance of cleaner evidence. We agree that there are some competing hypotheses that we cannot rule out completely due to data limitations; however, as we highlight in some of the comments below it is important to recognize that a viable competing hypothesis must be able to account for both the reduction in the GM yield advantage and the subsequent rebound. If improved breeding toward cultivars with GM traits was the sole driver, then how did the conventional cultivars catch up? If pest suppression was the sole driver, then why did the eroded gains subsequently rebound? Furthermore, if both these events did indeed happen, why did they happen coincidental to the timing of widespread pest resilience and the introduction of pyramided hybrids?

This is not to put the ball in your court, indeed it's our responsibility to convince the reader and we appreciate the engagement on this front. We tried very hard to balance the messaging around our resistance finding in the Discussion/Conclusion section to reflect that our results are “consistent with” resistance rather than “proof of” resistance. Please let us know if we did not achieve this goal.

Lines 237 to 238 state what I think is going on. The authors admit that their results may be due to “improved germplasm targeted in cultivars with stacked traits”. Companies charge more for maize hybrids with GM traits compared to those that do not have GM traits. Therefore, it is in the company interest to increase breeding effort toward these hybrids that they can charge more for. This has been documented in the US and is thought to be one of the reasons why growers do not plant the required 20% non-Bt refuge.

That's a good point and is in line with the profit maximizing goal of the company. However, if the best germplasm was always targeted toward the GM hybrids then why did we see a big reduction in their yield advantage prior to the introduction of pyramided cultivars? Companies that are

trying to build a brand reputation have an incentive to keep that premium stable; any uncertainty in the brand's reputation will hurt their bottom line. Especially in the SA context where GM adoption is a contentious issue. We have included some verbiage in lines 252-255 that discusses the high bar that a competing hypothesis would have to overcome, but would be happy to include some additional verbiage about profit-maxing objectives of the companies if you think that will help.

The description in lines 358 to 363 and portrayed in figure SA11 is an attempt to sidestep this argument, although it's not overtly stated. The problem is that we still don't know how much breeding effort for yield gain went into Bt or non-Bt hybrids. There is no detail on what the hybrids are. In the US, both Bt and non-Bt hybrids developed from larger seed companies (Dekalb and Pioneer, for example) are often sold to smaller seed companies and re-labeled for sale. Were the hybrids that were tested in this study simply relabeled from year to year or were they unique? There isn't enough detail for me to judge this.

This is a good point which should have been better addressed in the original manuscript. Each seed company needs to apply through the registrar to have a synonym listed on the national variety list. When a company applies for variety listing, they must provide information on where the material comes from and how the variety was bred. Upon application and submission of the seed sample, the variety is compared by physically planting it out with other listed varieties of the same class/maturity/grouping etc. The UPOV DUS system is followed to ensure that the variety is Distinct (new), Uniform, and Stable. Plant Breeders Rights are always considered, and this legal framework ensures that IP of breeders cannot be exploited.

As such, varietal names stay consistent from year to year. All new varieties must comply with the following requirements of the so called DUS test (D = Distinct, U = Uniform and S = Stable), before it can be approved for listing in varietal lists: it must have an acceptable denomination, in other words, it must not be the same as that of existing varieties, must not consist solely of descriptive words. According to UPOV rules the name of a variety may not change unless certain words have an unacceptable meaning in another language or may be difficult to pronounce in another country. The objective is that a specific variety must always be identified by the same name in all countries.

Further supporting this, the explanation that reports of field resistance timing (line 241) with a decline in Bt efficacy are unconvincing without any measurement of insect feeding in non-Bt or Bt hybrids. The authors report "widespread resistance" by 2007-2009 and the introduction of Cry1A.105 + Cry2Ab2 in 2012/2013, and the yield upswing in Bt versus non-Bt, as well as single Bt versus pyramided Bt's (Figs. 2-4) make this plausible. However, pest suppression could explain the difference in yield rather than resistance. As more and more Bt maize was planted in South Africa, populations could be suppressed by Bt. This would mean that the Bt yield advantage would decrease over time relative to the non-Bt hybrids. The yield upswing could be due to many other factors- improved breeding for yield in Bt hybrids, the introduction of an invasive insect, natural pest population increases, etc.

From 2009-2018, the percentage of conventional hybrids is not changing substantially, nor is the total percentage of Bt hybrids. However, the relative percentage of stacked vs single-Bt changes

from majority single-Bt to majority stacked. Further, if pest suppression followed by pest population increases were the underlying cause, we would not see an advantage of the stacked hybrids over conventional and single-HT but an eroded advantage of the single-Bt hybrids.

While the majority of field surveys report presence of resistance during the time periods/regions of interest, a few also report pest incidence. A survey of 30 farms in the Vaalharts region in 2008/09, where resistance was first reported, showed a mean incidence of 12.6% on Bt fields and 14.0% on non-Bt refuges. In 2009/2010, a survey of 7 farms revealed similar levels of pests. In 2011, a sample of 41 Bt fields and 39 non-Bt refuges found stem borer in 41% and 72% of locations, respectively (Van den Berg et al., 2013).

Another study which surveyed 105 farmers in 6 towns across 3 districts showed high incidence of stem borers on Bt fields in 2008/2009 and 2009/2010 in the town surveyed in the North West province, with at least 60% of farmers indicating an infestation level above 10% in both time points. Further, incidence levels decreased as the distance from the Vaalharts region increased for the time points surveyed. This corresponds to the beginning of the erosion of yield premiums in our results from the North West province, though these survey results do not report pest pressure for non-Bt fields (Kruger et al., 2011).

While these results comprise small samples, they indicate the presence of pests during periods/regions of interest and provide evidence that lack of pest pressure is not the cause of yield premium erosion.

However, we agree that there are other potential factors, including improved breeding for stacked Bt over single Bt, single HT, and conventional hybrids, which we discuss in lines 246-260. Release year may act as a proxy for genetic gain, but we cannot distinguish breeding efforts among hybrid types.

The lack of pest population measurements is concerning for another reason. Let's accept the premise that the results are due to resistance. Often when resistance evolves, the resistance is localized across the landscape, rather than being evenly spread across the landscape. The approach that the authors used could miss this effect if the trials were located in areas where resistance was either high or low. Line 369 states that the model included location by year as a component to eliminate pest pressure. But it's unclear whether location could have had outsized effects that the statistical model might not have removed.

Good point, thank you for bringing it up. While we remain optimistic that the inclusion of location-by-year fixed effects will implicitly control for changes in pest pressure across both space and time, we appreciate your concern. Since there is no available data on pest pressure at the scale of this study all we can do is look for anecdotal evidence that our model has not appropriately controlled for confounders.

Any competing hypothesis would have to explain why it caused the GM effect to reduce for BOTH single Bt and stacked trait while only bouncing back for the stacked trait (See figure 4). That is a tall order and we can't think of one that better fits the pattern of results. Maybe there was a strategic initiative among breeders to enhance the genetics of the conventional lines thus allowing

it to “catch up”, which would explain the reduction in the GM effect. But then how could that same initiative explain the subsequent increase that was only experienced by stacked traits?

We hope that our discussion of this caveat in the Discussion/Conclusion section is acceptable, we are trying to be fair and open about the possibility of competing hypotheses while also noting that they have a high bar to get over. We are happy to consider any specific changes you feel are in order.

Finally, as mentioned previously, there is too little information to make this experiment repeatable. For example, the only detail we are given on the field designs is from lines 332 to 347. This explanation is not sufficient. The authors refer to “cultivars”, which makes me wonder if all the non-Bt cultivars were hybrids, rather than open pollinated varieties. It would not be fair to compare these without specifying this.

Thank you for this comment. We have added additional information about the trials in the Methods section on lines 358-361:

“Seed dressing is widely used in South Africa and in the trial data. The same rate is applied regardless of the technology throughout the trials.”

And

“All cultivars in the trials are hybrids.”

As another example of how detail is important, in the US, all field corn hybrids are overtreated with a seed dressing of insecticide prior to sale. However, the rates of these insecticides tend to be higher on Bt hybrids compared to non-Bt hybrids. Because of this, any potential yield differences could be due to the soil insect pest complex and not Bt.

Yes, seed dressing is widely used in South Africa and in the trial data. The same rate is applied regardless of the technology throughout the trials. The only difference in the dosage is between dryland and irrigated maize, where irrigation maize receives a higher dosage. That being said, our study only focuses on dryland maize, and as such, each cultivar (regardless of technology) uses the same treatment. We now include this information in the Methods Section, lines 358-361.

A couple specific questions:

Line 138- is this what lines 405 and 406 are describing. Different “release” points by year? What is the justification for breaking at these points?

The subsamples in Figure SA1, which reports the associated estimates, are 2005-2007; 2008-2012; and 2013-2018. The first break was chosen based on when widespread resistance was being reported and the second break is when the pyramided event was released. The estimates for each split sample are from the baseline model, equation (1) in the Methods, and we only include stacked cultivars that were released into the trials during those years.

Essentially, we are trying to motivate our empirical strategy of interacting the GM effect with release year. By subsampling we are able to show that there was not only a decrease, which was shown in Figure 2, but also a subsequent rebound among stacked cultivars which was not evident in Figure 2.

Then, since subsampling is admittedly arbitrary, we move to the main focus of our empirical strategy which uses the full sample but interacts the GM effect with release year as specified in equation (2).

To make this clearer we have added clarifying sentences around the breakpoints on lines 138-141. We have also included verbiage clarifying our main (preferred) model on lines 178-182.

104 locations (line 312) times 36 years is a ton of degrees of freedom. I would suggest a statistician look at this. Were locations combined?

We appreciate this concern. This is a fairly typical fixed-effects statistical approach for estimating regression models with panel data that contains a large amount of unobserved heterogeneity. Another reviewer has picked up on some statistical issues that we have addressed.

Reviewer #2 (Remarks to the Author):

The authors have gathered more than 30 years of field-trial data from across South Africa to address the question of whether the effect of genetically-modified (GM) cultivars on yield changes over time and, if so, how. Three different categories of GM cultivars are considered: the soil bacterium *Bacillus thuringiensis* (Bt) cultivars, herbicide-tolerant (HT) cultivars, and cultivars that are both Bt and HT. The yields of the GM cultivars were compared to that of conventional non-GM cultivars. Various models were developed to quantify the effect of GM cultivars on yield. This review focuses on the statistics used in the analysis.

Thank you very much for the feedback, you raise some very important points that we address below and the manuscript is much improved as a result. Please note that we are providing a marked-up version of the manuscript that tracked all changes as well as a “clean” version that accepts all those changes. The line references in the replies below apply to the clean version.

The overall design of the study needs clarification. The authors specify that a “randomized complete block design (RCBD) with multiple replications was used throughout and each locality was allocated its trial randomization” (lines 328-329). What is meant by “multiple replications”? Is it the number of blocks within an RCBD at a given location, the number of RCBDs within a location, the number of RCBDs across locations, or the number of RCBDs both within and across locations? This would help the reader understand what a replicate number represents and perhaps why it had no significant effect. If the replicates are RCBDs within a location, was the randomization the same for all RCBDs?

This is a good question which should have been more thoroughly addressed in the original manuscript. A Randomised Complete Block Design with three replicates to accommodate the 50 genotypes was used throughout. Each locality was allocated its own trial randomisation that differs annually. The same 50 genotypes were used in all the trials and consisted of genotypes entered by the seed companies involved. Seed company entries are in order of priority. Where too many entries were received the final genotype choice was made through negotiations with each seed company. Stands of 40 plants per netto plot were prescribed throughout whereas the choice of plant population (plants/ha), row widths and spacing was left to the discretion of the co-workers. A number of 22 plants in each of two plant rows per gross plot were recommended for row widths less than 1.5 m, while a number of 42 plants in a single plant row per gross plot was recommended for row widths of 1.5 m and greater. Border rows were only prescribed for the perimeter of the trial and no border rows were required between plots. One boundary plant had to be removed from each end of the plant row at harvest.

We have now enhanced the discussion around this point on lines 343-361 in the Methods Section

How was y_{ilt} , the yield for cultivar i at field-trial location l in year t , computed? Was it the yield for each plot, multiple observations for each cultivar within the RCBD, the least squares mean from the individual RCBD analyses, or the average of the observed cultivar i yields? (These last two differ if any missing values are present within the RCBD.)

Apologies for the notational oversight, data observations are at the replicate level. For clarity, we now note specifically under equation (1) that the sample included these replicates, lines 407-408.

In general, one should determine whether an effect is fixed or random and let that determine the model fit instead of fitting effects as both fixed and random. Was a mixed effects model fit? That would require either the intercept or the GMs to be fixed and the other random; otherwise, if both are random, it is a random effects model. Note: The slopes are not random for the mixed effects model; the GMs are random. Given the study design, the assumption that GMs are random seems most reasonable.

Thank you for the note, apologies for the confusion. Equations 1 and 2 are fixed effects models based on econometric terminology. They are estimated using OLS and the location-by-year fixed effects are essentially separate intercepts for each location-year group. These are the models for our main results reported in Figure 4. Note in the fixed effects econometric framework the terms “fixed” and “random” are used differently than in a mixed effects model, essentially denoting different assumptions about the correlation of the factor variables with the error term. Andrew Gelman’s blog post below has a great discussion in the comments on this issue:

https://statmodeling.stat.columbia.edu/2010/12/17/so-called_fixed/

Both fixed effects and mixed effects models are reasonable approaches here. We opted for the former but also included the latter as a robustness check whereby we include location and year as random effects, and random slopes are also included for the GM effect across three dimensions: release year, location, and year. As discussed on lines 216-221, the results are consistent across both approaches.

For the primary modeling effort, the three different types of GM cultivars are grouped together to determine the impact of GM cultivars on yield. The effect of the GM cultivar differs among the Bt, HT, and both Bt and HT cultivars. The factorial nature of the cultivar types could be reflected by considering Bt and HT as indicator variables with a value of 1 if, respectively, Bt or HT is present. For the conventional cultivar, both indicator variables would be 0. By collapsing them into one category, the effect of GM depends on the effect of each of the three GM types and the proportion of times each occurs in the dataset for a specific year affects the estimates. Thus, a change in the effect of GM could be simply due to a higher proportion of one type of the GM cultivars occurring in the test data in a given year and that proportion may not be representative of the population of interest (maize grown in South Africa).

We appreciate the concern, but its not clear to us whether this is impacting our results. Figure 4 summarizes our main findings where we sequentially go through pairwise comparisons of a particular group versus another. Our goal with this approach is to try to tease out a likely mechanism for the observed reduction and subsequent increase in GM yield advantage.

Each panel of Figure 4 is a result of estimating equation (2) using a subsample of the data, eg for Sing Bt v Conv we only keep cultivars that are either single trait Bt or conventional and the latter is used as the baseline. When we estimate Sing Bt v Sing HT then single trait HT is the baseline.

This is mentioned on lines 171-176 in the manuscript. We have also added new verbiage clarifying the modeling approach on lines 178-182.

Regarding the number of observations by groups, our belief is that this is only a concern when one of the groups has a very small number of observations relative to the others. We also include location by year fixed effects, so that the only information driving the estimate is yield variation within each trial.

To make sure this wasn't a concern, we also replicated figure 4 when we double the number of observations for each of the GM groups. For example, when comparing single Bt to conventional, we take all the single Bt observations and add them to the sample to double their number of observations. The figure below reports these results, which are very similar to figure 4 in the manuscript and thus we conclude this isn't a major concern.

Replication of Figure 4 when Observations from GM groups are doubled:

The authors have considered several follow-on analyses. To the extent possible, it would be better to have an over-arching model that would allow the desired comparisons. As an example, all types of GM cultivars could be part of the initial model. Another example would be to include a categorical variable for province and its interaction with GM.

Thank for this suggestion, we see your point but respectfully disagree. To the extent that we have a single overarching model, it is given by equation (2) in the methods. This is the model used to estimate our main findings which are reported in Figure 4, where each panel reports estimated yield gains for each of four subsamples. By first subsampling the data and then estimating the pairwise relationship, the location-by-year fixed effects are unique to that pairing. Since these are the main control variables for our identification approach this represents a more robust approach than pooling all the data together and estimating equation (2) with different indicators for the GM categories.

This same logic applies to the province-specific approach; by splitting the sample and then estimating the effects we are leveraging a more robust model.

It is also worth noting that we do compare our results to a mixed model, which is arguably the best possible single (overarching) model approach one could consider in this context, and the results are consistent with our preferred approach.

Thank you very much for the opportunity to clarify this, we now include this discussion when we introduce equation (2) in the Methods section on lines 433-443.

A large number of tests are being conducted. Some correction should be made as the type I error rate is not well controlled as it stands. Prediction bands, which are a little wider than confidence bands, for the fitted curves are more appropriate than showing individual confidence limits as in Figures 3 and 4. Since the curve is significantly different from $y=0$ and the prediction bands would probably lie fully above the zero line, one could produce an estimate of the effect of GM for any point in time. The model becomes more complex, but a deeper understanding of the system is gained.

Thank you for the opportunity to clarify this. As mentioned above equation (2) is the preferred model and it has three parameters capturing the dynamic GM effect. The estimates of these parameters drives the shape of the main results reported in Figure 4. The parameter estimates are reported in Table SA2 and the p-values for a joint test of statistical significance are less than 0.000 for all but the single-Bt vs single-HT comparison.

We now include this verbiage on lines 178-182 in the manuscript. We hope this helps clarify things, we do not think adjustments for multiple comparisons are necessary here.

The authors have undoubtedly spent an enormous amount of time obtaining, cleaning, and analyzing these data. The results are potentially interesting. It would help the reader if some of the statistical work, especially the design, was better described and more care was taken with some statistics.

Thank you very much for your feedback, it really helped us recognize where the communication needed improvement. We stand ready to address any remaining concerns that you have.

Reviewer #3 (Remarks to the Author):

This is a very interesting manuscript with some high quality data and an important story to tell to a wide audience.

To summarise the key results these are:

1. Bt maize cultivars led to yield increases compared to conventional cultivars
2. the yield advantage started to disappear once insect populations started to develop resistance to the single Bt gene (protein)
3. the yield advantage was reinstated (and possibly better) when cultivars with two pyramided (and crucially different to the first single Bt Cry1Ab gene and protein)

However there are some significant improvements to be made to include the clarity of the interpretation for the reader.

Thank you very much for the feedback, you raise some very important points that we address below and the manuscript is much improved as a result. Please note that we are providing a marked-up version of the manuscript that tracked all changes as well as a "clean" version that accepts all those changes. The line references in the replies below apply to the clean version.

I do not believe the title is very clear in summarising the findings. It does seem to imply that Bt led to yield losses, whereas this was not the case.

Perhaps more engaging titles could be:

Two gene Bt maize cultivars were more resilient than single gene Bt cultivars in South Africa

or

Single gene Bt maize cultivars lacked resilience to insect losses in South Africa

or

Improving the sustainability of maize yield with two gene Bt cultivars in South Africa

Interesting thought. We agree that these titles are likely to be more engaging, but based on some of the feedback from the other reviewers these alternatives might be seen as overselling the paper. We are open to changing it at the Editor's discretion.

Indeed the Abstract has some lack of clarity and should explicitly state that single gene Bt cultivars were released initially and failed because of "resistance in the insect populations". Indeed perhaps it could be concluded that the race to deploy single gene Bt was probably a mistake. And the deployment of 2 gene Bt also necessitated the use of 2 new and different genes. Indeed this had been previously widely predicted (for example see Zhao et 2005, PNAS - a team from Cornell).

Thank you very much for this citation, very interesting and relevant paper. We now use it as our closing discussion to highlight the importance of a holistic risk management program, see lines 305-314.

We have also modified the abstract to focus the resistance verbiage on the single gene Bt cultivars.

Another issue that is not discussed is the returns to farmers. Higher yields would increase farm profitability. In addition, if insecticides were being used to control insects in conventional fields this would mean a higher input cost and overall a considerably lower profit margin.

This may not be easy for the authors to provide data but there could be some consideration as to the relative losses of maize to different insect pests and how that changed over time eg from *Busseola* to FAW? Did FAW have a large impact on conventional yields.

Thank you for this good point. While profitability is ultimately the driver of adoption of any genetic material, it is outside the scope of this study. The robust yield data set used in this study lacked insecticide application rates (besides denoting that best management practices were used) and cost of applications. Given the heterogeneity in application costs (boom size, efficiency, number of applications, insecticide costs, etc.) any inclusion of costs would simply be conjecture. This is not to downplay the importance of the economics of the situation, but rather to say that the findings we present are a function of robust yield modeling rather than wading into the pool of a wide range of cost assumptions which would be necessary to provide estimates for this excellent question.

There are also some low yielding years in the mix (eg 2006) - was this due to insect pressure or a biotic stress?

Thank you for highlighting this. South Africa, specifically the maize growing region in 2006, received above-average rainfall during January, February, and March due to La Niña patterns. Coupled with the March 2006 temperatures being well below-average reduced heat units and slowed crop development in the grain-filling stages reducing yield with spatial heterogeneity, ultimately resulting in lower than average yields for the country.

This highlights an important component of our approach. Unfortunately, it is not common practice for analyses of GM impacts on yields to control for weather effects before identifying the yield premium, which can bias estimates downward if not accounted for. Here we use location by year fixed effects which will account for the variation in yields across trials and thus focus more directly on within-trial yield differences.

Some minor comments"

line 18 "less insecticide application" should be "fewer"

Corrected, thank you.

line 219-223 should be explicit that the loss of over 35 million food rations was due to poor resistance management and not Bt deployment per se

Great point, we have added the sentence “Importantly, these loss estimates were not due to the use of GM but rather the erosion of a yield premium relative to conventional maize due to resistance” to clarify this messaging in the abstract.

Supp Figure SA11 - what is the significance of the horizontal red line on each graph

The horizontal line reports the average release year for each group of cultivars. Upon reflection this is not a very useful statistic and has been removed from the figure. Thank you for noting this.

Reviewer #1 (Remarks to the Author):

This is my second time reviewing this manuscript. I appreciate the author's thoughtful responses to my concern, especially regarding pest suppression. However, I still think the authors are putting too much confidence in pest resistance as an explanation rather than breeding.

Thank you very much for the previous feedback and the ongoing engagement. I think we share a similar sentiment that the study is not 100% conclusive and there are a myriad of potential alternative hypotheses. We feel that we have made every effort to place the paper appropriately, highlighting this possibility while also ruling out a large number of competing hypotheses.

Figure 4 could simply tell a breeding story. Let's assume that companies put most of their breeding efforts into their most "elite" hybrids that they can charge a premium price for. In that case, corn breeders released hybrids with single Bt traits, with a yield advantage due to breeding that eroded over time. This yield advantage was apparent compared to both conventional corn hybrids (Fig. 4a) and single HT corn hybrids (Fig. 4c).

Breeders then stopped putting so much effort into these single Bt trait hybrids when stacked Bt trait hybrids were released (Figs. 4b and 4d). Why the upswing in yield? Breeding could explain it all to me.

We appreciate this counter example and are thankful that you brought it up. It does explain the downturn in the single Bt lines and also the upswing in the stacked ones; however, it does not explain why the stacked gain (Fig 4b) eroded ~66% between 2006-2013 (from ~.6 to ~.2). Thus it does not seem consistent with the results.

We highlight this challenge for competing hypotheses in the discussion around line 252, noting "While we cannot directly test this resistance hypothesis due to a lack of pest population data at the scale of this study, any competing hypothesis would have to explain why the GM yield premium reduced for both single Bt and stacked trait cultivars while only bouncing back for the stacked trait cultivars.

I'm not refuting the author's claim that we can explain the dips and upswings in yield could be due to resistance. I only think they should make the point that breeding is a good alternative hypothesis that's not mutually exclusive.

We really appreciate the constructive pushback and feel the paper is much better as a result. Please note that we do devote an entire paragraph to this in the Discussion (around the referenced line above), we are trying to be very open-minded about the implications of the analysis and stand ready to make any specific changes to the verbiage where warranted.

Reviewer #2 (Remarks to the Author):

As Reviewer 2 for the original manuscript, this reviewer will continue to focus on the statistics used in the analysis.

We have very much enjoyed the constructive feedback and are thankful for your time and effort in reviewing the manuscript.

First, the authors have shown careful consideration of each of the comments, at times revising the manuscript and, at other times, explaining why they did not make changes in response to a comment. Thank you for making that effort. In this version, it is evident that the results are robust to the choice of model, which adds confidence to the conclusions being drawn.

Thank you for helping us enhance the manuscript, we feel it is much stronger as a result.

Apparently, one comment in the original review was not fully explained. It is the following:

“For the primary modeling effort, the three different types of GM cultivars are grouped together to determine the impact of GM cultivars on yield. The effect of the GM cultivar differs among the Bt, HT, and both Bt and HT cultivars. The factorial nature of the cultivar types could be reflected by considering Bt and HT as indicator variables with a value of 1 if, respectively, Bt or HT is present. For the conventional cultivar, both indicator variables would be 0. By collapsing them into one category, the effect of GM depends on the effect of each of the three GM types and the proportion of times each occurs in the dataset for a specific year affects the estimates. Thus, a change in the effect of GM could be simply due to a higher proportion of one type of the GM cultivars occurring in the test data in a given year and that proportion may not be representative of the population of interest (maize grown in South Africa).”

The authors assumed that the above comment was referring to Figure 4. In fact, Figure 4 illustrates that GM and time interact; that is, the differences among the different types of GM (single-trait Bt, single-trait Ht and stacked trait) change over time. Figure 3 depicts the main effect of GM, which is an estimated average of yield gains of GMs released in a given year and in the study; it is not clear whether all new releases are included each year. Because of the interaction, the yield gain for each year is not representative of any of the types of GMs when more than one type is present (see Figure 4) and is heavily dependent on the cultivars included in the study, which may not be reflective of the cultivars used by farmers. Although the authors note that the model and resulting Figure 3 are based on the assumption of “a homogeneous effect across different types of GM cultivars” and then explore the assumption, their interpretation appears to draw inference to the broader population, which is not homogeneous. In this case, the proportion of cultivars of each type included in the study has a substantial impact on the estimated yield gain simply because the yield gain varies with type. Thus, the recommendation was made to consider a model with type of GM (conventional (no GM), single-trait Bt, single-trait Ht and stacked trait) instead of only considering presence/absence of some type of GM in model 2. Due to interaction, this would lead to separate curves for the yield gain from each type of GM in Figure 3 and would eliminate the need for Figure 4.

Thank you for this, I think we can now see where the miscommunication was happening on our end. In the previous comment we took “For the primary modelling effort” to refer to Figure 4 since that is the model that we base our results on. We view Figures 2 and 3 as motivation for a more detailed approach that looks at specific traits as in Figure 4, whereas the comment suggested amending the model in Fig 3 to include trait-specific effects and interactions with time. I think we ultimately end up at the same place that you suggest going from a modeling perspective, we just take a little extra time getting there.

We would be happy to remove Figure 3 altogether at the Editor’s discretion, but our preference is to keep it as it helps illustrate that there is something interesting going on in the data that requires deeper insight.

Below are some further suggestions, none of which are major:

1. Beginning at the end of line 164, “the yield effect” should be “the GM effect”

Corrected, thank you.

2. Beginning at the end of line 181, “less than 0.000” should be “less than 0.001” (It is not possible to have a p-value less than 0.)

Corrected, thank you.

3. Line 343: Since there are three blocks (replicates) for each RCBD and a new randomization was conducted for each year-location combination, clarity could be added by changing “with multiple replications was used throughout and each locality was allocated its trial randomization” to “with three replications (blocks) was used throughout and each locality was allocated a new trial randomization each year”

Corrected, thank you.

4. For all graphs of yield gains (most graphs), the y-axis was labeled as “Yield (MT/ha).” This should be changed to “Yield Gain (MT/ha).”

Corrected both in the main figures as well as the supplementary, thank you.

5. In Figure 4, the title indicates that plot (d) depicts the yield gain of single-trait HT vs conventional, but the title above the plot says that it is the yield gain of stacked trait vs single-trait HT. This should be clarified.

Corrected, thank you. The correct verbiage is “(d) compares stacked cultivars to single-trait herbicide tolerant (HT) cultivars”.

Again, the authors have conducted a thoughtful analysis of a large, interesting dataset.

Thank you again for the engagement.

Reviewer #2 (Remarks to the Author):

The comments in this document are from the Editor based on (in part or addition to) feedback from Reviewer 2. Comments are in plain text, with replies in *italics*.

In your revision, please address the remaining comments of Reviewer #2. We hope that the new figure and analysis generated in response to our correspondence will assist in this revision, although we are concerned that there are still some discrepancies remaining that will need to be resolved, as explained in more detail below.

Thank you very much for the invitation to resubmit the manuscript. We have updated the manuscript to include the new model that pools the data and estimates separate effects of GE across traits. Alongside the main submission, we have also included a “tracked” version so that you can see where all the changes in the main text have occurred.

As noted by Reviewer #2, we also believe that the current Figure 3 should be removed. It appears from the text and other figures that the time series for “single BT”, “single HT” and “stacked” begin at different years, and so the line in the current Figure 3 is in fact representing different things in different years, which could cause misunderstandings by readers.

We have replaced Figure 3 with a new version that adds the trait-specific dynamic effects. The parameter estimates for this new model are reported in column M3 of Table SA2. We have also added discussion of the new model in the main text when presenting Figure 3.

We also agree with Reviewer #2 that the original analysis should ideally have included the GM type as a factor. Therefore, we appreciate you performing the new analysis and generating the new figure that shows the separate lines for “single BT”, “single HT” and “stacked”. However, we are still uncertain about the following:

(1). The new statistics table provided appears to show that only the “stacked” GM type has a significant interaction with time, in contrast to the claim given alongside the new figure that all three trends were significant. Did you perhaps mean that all three GM types had a significant effect, but only the “stacked” type had a significant temporal trend? Please make sure that it is transparent in the text and figure legend(s) which temporal trends are statistically significant.

Table SA2 reports joint hypothesis tests of the release-year interactions for each of the trait groups (see rows reporting “P-value of joint release year effect”). They are all statistically significantly different from zero at standard significance levels, i.e. 0.05.

(2). We had understood that the “single HT” cultivars only appeared in 2003 and the “stacked” cultivars only appeared in 2005. Therefore, we are uncertain why all three lines in the new figure begin at year 2000.

The previous figure that we shared with the Editor has been updated to reflect when each trait group was released.